# HalOmi: A Manually Annotated Benchmark for Multilingual Hallucination and Omission Detection in Machine Translation

**David Dale, Elena Voita, Janice Lam, Prangthip Hansanti, Christophe Ropers,**
**Elahe Kalbassi, Cynthia Gao, Loïc Barrault, Marta R. Costa-jussà**

FAIR, Meta

{daviddale,lenavoita,janilam,prangthiphansanti,chrisropers
ekalbassi,cynthiagao,loicbarrault,costajussa}@meta.com

## Abstract

Hallucinations in machine translation are translations that contain information completely unrelated to the input. Omissions are translations that do not include some of the input information. While both cases tend to be catastrophic errors undermining user trust, annotated data with these types of pathologies is extremely scarce and is limited to a few high-resource languages. In this work, we release an annotated dataset for the hallucination and omission phenomena covering 18 translation directions with varying resource levels and scripts. Our annotation covers different levels of partial and full hallucinations as well as omissions both at the sentence and at the word level. Additionally, we revisit previous methods for hallucination and omission detection, show that conclusions made based on a single language pair largely do not hold for a large-scale evaluation, and establish new solid baselines.

## 1 Introduction

With neural machine translation systems reaching an overall satisfactory quality, alleviating those rare but severe translation pathologies that undermine user trust becomes very important. These pathologies include hallucinations (translations containing information completely unrelated to the input) and omissions (translations that do not include some of the information of the input). While understanding hallucinations is receiving increasing attention (Raunak et al., 2021; Müller and Sennrich, 2021; Zhou et al., 2021; Guerreiro et al., 2023; Dale et al., 2023; Guerreiro et al., 2022), progress in this direction is hampered by the lack of annotated data. To the best of our knowledge, previous datasets are limited to German-English data with sentence-level annotations of hallucinations and omissions (Guerreiro et al., 2023) and Chinese-English data with token-level hallucination labels (Zhou et al., 2021). Previously available general-purpose quality assessments, such as direct assessment (DA) Gra-

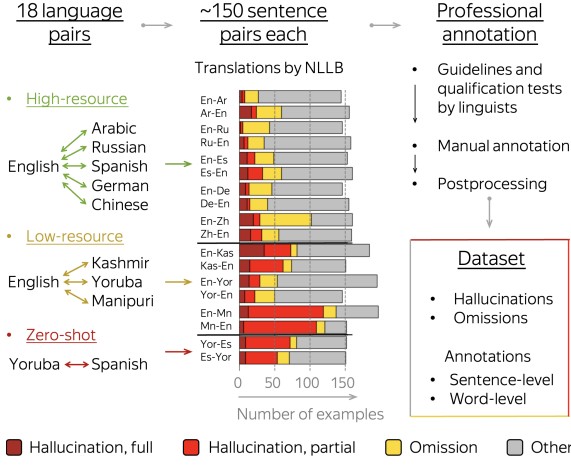

Figure 1: Dataset summary.

ham et al. (2013), MQM (Lommel et al., 2014), or XSTS (Licht et al., 2022) do not seem suitable since they do not distinguish between hallucinations, omissions and other translation errors. In this work, we aim to address this limitation.

Ideally, an evaluation dataset for hallucination/omission detection should satisfy several conditions: (i) data has to cover a broad range of languages with varying resource levels and scripts, (ii) translations should be generated naturally, (iii) the models that produced the translations have to be available, and (iv) considered modeling paradigms have to cover several approaches (i.e., encoder-decoder vs decoder-only, single language pair vs multilingual, etc.). The first point is important because the best-performing detectors are different for high- and low-resource settings, and general conclusions cannot be made based on a single language pair (Section 4). Secondly, translations have to be generated naturally as opposed to using specifically developed perturbations of models and/or data because conclusions for the latter might not transfer for detection of natural hallucinations (Section 6). Thirdly, the corresponding model should be released along with the translations to allow evaluating "internal" detection methods. Fi-

nally, in the most ideal setting, various models are needed to test whether the detection methods transfer between modeling approaches.

While satisfying all the desiderata is very challenging, we can satisfy all but last by focusing on the state-of-the-art multilingual NLLB-200 model (NLLB Team et al., 2022). In addition to covering a broad range of languages and being publicly available along with its training data, NLLB is widely recognized[1] and is likely to stay the state-of-the-art for the foreseeable future. For this model, we choose 18 language pairs that include high- and low-resource languages, as well as a zero-shot pair (Figure 1). We develop rigorous annotation guidelines for identifying full and partial hallucinations and omissions and use these guidelines for manual annotation of translations in all 18 directions. The resulting dataset contains fine-grained sentence-level and token-level annotations.

We highlight the importance of our dataset by making several valuable observations that would not be possible otherwise. For example, we find that for low-resource directions, internal methods perform much better than external methods that substantially fail. When analyzing performance of a range of recently introduced pathology detection methods, we see that some of the previous results do not transfer across languages. As another example, we show that relying on attention to make conclusions about translation quality is very fragile. Finally, we introduce some detection tasks (e.g., token-level omission detection) that became possible only with our data. We believe our work opens the door for reliable and accessible research on detecting and analyzing translation pathologies as well as understanding their causes.

Overall, we:

- release a dataset with fine-grained professional annotations of hallucinations and omissions for 18 language pairs[2];

- analyze previous sentence-level detectors and find that e.g. (i) for low-resource settings, model internal characteristics work best,

(ii) attention is very fragile when used to judge translation quality, among other observations;

- introduce word-level pathology detection tasks along with the baselines.

## 2 Dataset Creation

The steps to create the dataset were (i) choosing the language pairs, (ii) gathering data for annotation, (iii) developing annotation guidelines and qualification sets, (iv) manual annotation, (v) post-processing. Here, we explain these steps.

### 2.1 Selection of Languages

We optimized the language selection in order to cover (i) different resource levels and (ii) a variety of language families and scripts. Among the languages available in NLLB-200, we include 5 high-resource language pairs (Arabic, Mandarin Chinese, German, Russian, and Spanish paired with English), 3 low-resource language pairs (Kashmiri, Manipuri, and Yoruba paired with English) and a zero-shot pair (Spanish-Yoruba).[3] We consider all language pairs in both directions which gives us 18 translation directions summarized in Figure 1.

### 2.2 Gathering Data for Annotation

Since strong NLLB models rarely generate hallucinations and omissions, getting translations that are likely to contain these types of errors is not straightforward. To gather these translations, we developed a multi-step procedure where we first choose data to generate translations and then choose a subset of the generated translations for annotation.

**Choosing data for translation.** Since we expect that the NLLB model will not hallucinate much when handling high-resource languages, in addition to clean in-domain data, we use noisier out-of-domain sources. Overall, the data we use to generate translations is as follows:

- *in-domain*: FLORES-200 development set (NLLB Team et al., 2022);

- *out-of-domain*: Jigsaw toxicity detection competition corpora (Jigsaw, 2020)[4] – for English, Russian and Spanish; comments from Wikipedia discussion pages[5] – for Chinese,

---

[1]Only 4-months after launching NLLB-200, Wikimedia reported that this was the third most used machine translation engine accounting for 3.8% of all published translations. Scientific impact is also prominent: the model has been used as standard to compare with other MT paradigms such as prompting with large language models (Zhu et al., 2023).

[2]The data and code are available at https://github.com/facebookresearch/stopes/tree/main/demo/halomi

[3]In the NLLB training dataset, Spanish and Yoruba sentences were paired to English but not to each other.

[4]https://www.kaggle.com/competitions/jigsaw-multilingual-toxic-comment-classification/

[5]From public dumps: https://dumps.wikimedia.org/.

| Definitions | Hallucination vs mistranslation | Severity levels |
|---|---|---|
| • **Hallucinations**: contain information completely unrelated to the input.

• **Omissions**: some of the information contained in the source is not present in the translation. | To decide whether a source token "corresponds" to an erroneous target token, answer:

○ Does this source word fall into the common meaning category as this target word?

○ Does this source word have a semantic connection with this target word?

○ Can you try to come up with a reasonable theory on how this source word is associated with this target word?

If "no" to all, then hallucination. | Depending on the number of words that are hallucinated/omitted, a pathology can be

• **Word-level**: only 1-2 words,

• **Partial**: at least 3 words, but not all;

• **Full**: all except maybe 1-2 words.

The severity levels do not overlap (i.e., a partial hallucination is not full). |

Figure 2: Overview of some parts of the annotation guidelines.

Arabic and German. The Jigsaw corpora were extracted from Wikipedia talk pages, so the distributions of these texts are rather similar.

We translated these texts with the 600M distilled NLLB model[6] following the standard setting (beam size 5, forbidden generation of the <UNK> token, forbidden repetition of 4-grams, limiting the translation length to $3 \cdot \text{len(source)} + 5$ tokens.

**Choosing translations for annotation.** To find potentially pathological translations, we scored sentence pairs by multiple metrics that were used as hallucination detectors in previous works. Specifically, we used some methods from Guerreiro et al. (2023): ChrF++ (Popović, 2017), reference-based COMET[7] and referenceless COMET-QE (Rei et al., 2020), and Seq-Logprob (their best detector). We also used some methods introduced in Dale et al. (2023): cosine similarity coming from LASER3 (Heffernan et al., 2022) and LaBSE (Feng et al., 2022), a bidirectional XNLI score, and ALTI+ source contributions (Ferrando et al., 2022).

For each translation direction and data source, we selected sentence pairs with 3 strategies:

- Sample uniformly – to preserve data diversity and non-hallucinated examples;

- Sample favoring potentially pathological translations (with the probabilities proportional to the quantiles of the detectors);

- Pick the worst according to the detectors – to increase the chance of hallucinations.

Appendix A describes the amount of data selected by these strategies for all directions.

---

[6]We selected the smallest model from the NLLB family as the most popular one, and potentially the one generating most hallucinations and omissions.

[7]We applied the reference-based metrics only to the FLORES data.

## 2.3 Guidelines and Qualification Tests

To ensure annotation quality, guidelines and qualification tests were prepared by professional linguists.

**Annotation guidelines.** These guidelines define:

- the notion of hallucinations and omissions;

- the hallucination vs mistranslation distinction;

- hallucination/omission severity levels.

Figure 2 summarizes the resulting guidelines. Note that distinguishing hallucinations from other translation errors is one of the known difficulties when dealing with hallucinations (Raunak et al., 2021; Guerreiro et al., 2023). In our guidelines, a token is referred to as hallucinated if there is no corresponding token in the source (Figure 2).

For all pathologies, linguists provide positive and negative examples in diverse languages. Additionally, we ask the annotators to mark if a translation is incomprehensible, i.e. whether the text is garbled or in another language. These translations are then discarded.[8]

**Qualification tests and postprocessing.** For annotation, we choose professional translators (2 for each language) who are allowed to annotate our data only after passing a specifically developed qualification test. More details on this test and postprocessing steps can be found in Appendix A.

## 3 Dataset Description

**Annotation format.** The resulting data contains the source text and its translation, along with the

---

[8]We believe that incomprehensible texts should be considered separately for two reasons. From the user perspective, hallucinations and omissions are mostly fluent, which can mislead the user into trusting the translation; differently, incomprehensible texts are clearly bad sentences and thus do not mislead the user. From the detection perspective, incomprehensible sentences can be recognized regardless of the source, while hallucinations and omissions can be judged as such only in relation to the source sentence.

| | English-Arabic | English-Chinese | German-English |
|---|---|---|---|
| Input | Because of their success with submarines, after the war Germans aren't trusted to have many of them. | He had 2 goals and 2 assists in Washington's 5-3 win over the Atlanta Thrashers. | Selbstangabe) nur geprüfte Dokumente. |
| Output | سبب نجاحهم في استخدام الغواصات بعد الحرب، لا يُمكن الألمان من الحصول على الكثير من هذه الأغراض. | 他曾经在亚特兰大的比赛中获得了两项奖励,但他没有得到任何奖励. | It's just that I'm not sure what I'm saying. |
| Omission degree | Word-level | Partial | Full |
| Omitted word span annotation | Because of their success with submarines, after the war Germans aren't <<<trusted to>>> have many of them. | He had <<<2 goals and 2 assists in Washington's 5-3>>> win <<<over the>>> Atlanta <<<Thrashers>>>. | <<<Selbstangabe) nur geprüfte Dokumente>>>. |
| Hall. degree | No hallucination | Partial | Full |
| Hallucinated word span annotation | سبب نجاحهم في استخدام الغواصات بعد الحرب، لا يُمكن الألمان من الحصول على الكثير من هذه الأغراض. | 他曾经在亚特兰大的比赛中获得了两项奖励, <<<但他没有得到任何奖励>>>. | <<<It's just that I'm not sure what I'm saying>>>. |

Figure 3: Annotated examples from our dataset.

word-level and sentence-level annotations of omissions and hallucinations. Figure 3 shows examples of annotated translations from our dataset.

**Overall statistics.** Figure 1 shows the proportions of hallucinations and omissions in the data (translations with both hallucinations and omissions are referred to as hallucinations). Overall, all directions have at least 3% translations with hallucinations (1% full) and 17% with omissions (5% full). Most of the full hallucinations are also labelled as full omissions, and vice versa.

**Differences between resource levels.** From Figure 1 we see that, not surprisingly, high-resource language pairs hallucinate less than low-resource. A less anticipated difference between high- and low-resource settings is seen when looking within each language pair. In high-resource settings, translating to English leads to more hallucinations than translating from English. Differently, for low-resource pairs, translation from English has higher hallucinatory rates than translation to English for the same language pair. This might inspire future work to analyze the role of English data in the multilingual NLLB model. Finally, while for the zero-shot pair one might expect more pathologies, this is not what we see: results for the zero-shot pair are comparable to those for low-resource languages.

## 4 Sentence-Level Detection

Detecting pathologies at the sentence level is the task of flagging a whole translation as pathological or not. This is the standard definition of e.g. the hallucination detection task (Lee et al., 2019; Müller et al., 2020; Raunak et al., 2021; Guerreiro et al., 2023; Dale et al., 2023; Guerreiro et al., 2022; Xu et al., 2023). Such sentence-level pathology detection (instead of flagging individual erroneous tokens) is an integral part of hybrid pipelines when

a machine-generated translation is first passed to a quality estimation system and then, if needed, is corrected by human translators.

**Detection tasks.** For our dataset, we define three sentence-level detection tasks:

- *hallucination detection*: same as in previous work mentioned above;

- *omission detection*: detecting translations with omissions on a hallucination-free subset. The latter is to disentangle omissions from a more severe hallucination pathology;

- *pathology detection*: detecting translations that are either hallucinations or omissions.

**Evaluation methodology.** We evaluate the ability of a detector to rank more severe pathologies higher (e.g., full hallucinations higher than partial, any hallucinations higher than non-hallucinations, etc). For this, we use an adaptation of the binary ROC AUC score for several classes. Formally, we subtract from the perfect score, i.e. 1, the percentage of incorrectly ranked pairs of sentences with different labels. For two classes, this metric is equivalent to the ROC AUC score.

We compute the metrics for each translation direction separately.

### 4.1 Detection Methods

Detection metrics can be either internal, i.e. relying only on the information from the model that generated the inspected translation, or external, i.e. using external models. We use the best detectors from several recent works, along with some of their modifications we propose in this work. The metrics are summarized in Figure 4.

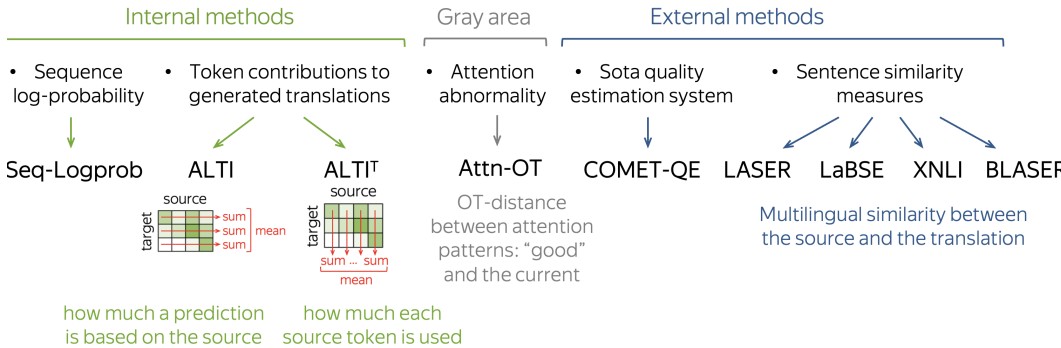

Figure 4: Summary of the sentence-level detection methods.

**Internal methods.** For internal models, we use the best method from Guerreiro et al. (2023) (sequence log-probability) and the best internal method from (Dale et al., 2023), ALTI. ALTI (Ferrando et al., 2022) is an attribution method that evaluates token contributions to generated translations. For hallucination detection, Dale et al. (2023) evaluate how, on average, the prediction of each target token is based on the source. Here, mostly to detect omissions, we propose a different variant ALTI$^T$ that computes how much, on average, each source token was used to generate the translation. Intuitively, if many source tokens are not used during generation, the translation is likely to not contain some information. The difference between the two versions of ALTI is illustrated in Figure 4.

**External methods.** For external methods, we use the state-of-the-art quality estimation system COMET-QE (Rei et al., 2020) and sentence similarity measures proposed in Dale et al. (2023). The latter are cosine similarities coming from LASER3 (Heffernan et al., 2022), LaBSE (Feng et al., 2022), and a bidirectional XNLI score. Finally, we evaluate a translation quality estimation method from Seamless Communication et al. (2023), BLASER 2.0-QE, built on top of SONAR sentence embeddings (Duquenne et al., 2023).

**Gray-area method.** Finally, we also use a recent optimal transport-based measure evaluating the abnormality of the attention distribution compared to those of good translations (Guerreiro et al., 2022). While this method uses internal characteristics, it requires external data, i.e. a large collection of attention maps for "good" translations, which can be hard to obtain for low-resource settings.[9]

---

[9]We use the best variant of the original method (Guerreiro et al., 2022). For more details, see Appendix B.

## 4.2 Experimental Results

The detection scores for hallucinations and omissions are shown in Figure 5. The scores for detecting all pathologies are given in Appendix C.

**High-resource: much easier to handle.** We can see that it is much easier to detect pathologies in high-resource settings: the gap between the best scores for high- and low-resource directions is rather big (e.g., for halucinations, 0.89 vs 0.79). Note also that for high-resource language pairs, both internal and external methods perform quite well (e.g., Seq-Logprob and LaBSE for hallucinations; XNLI, LaBSE and ALTI$^T$ for omissions).

**Low-resource: internal methods take the lead.** In low-resource settings, external methods drop substantially with some of them losing sensibility. For example, high-performing XNLI drops close to chance for all pathologies. Overall, most external models (with the exception of massively multilingual BLASER) are unlikely to be competent for low-resource directions as they do not observe enough relevant data during training. While previous work already expressed this concern and advocated focusing on internal methods (Dale et al., 2023), without our dataset, verifying this was not possible.

**Hallucinations: Seq-Logprob is the most stable.** After it turned out that the standard sequence log-probability is more informative for hallucination detection than the heuristics introduced earlier (Guerreiro et al., 2023), a few recent works reported improvements over Seq-Logprob: ALTI and LaBSE in Dale et al. (2023) and Attn-OT in Guerreiro et al. (2022). We see, however, that on average, Seq-Logprob is still the most robust accross translation directions. This discrepancy comes from the fact that those works made conclusions based on a sin-

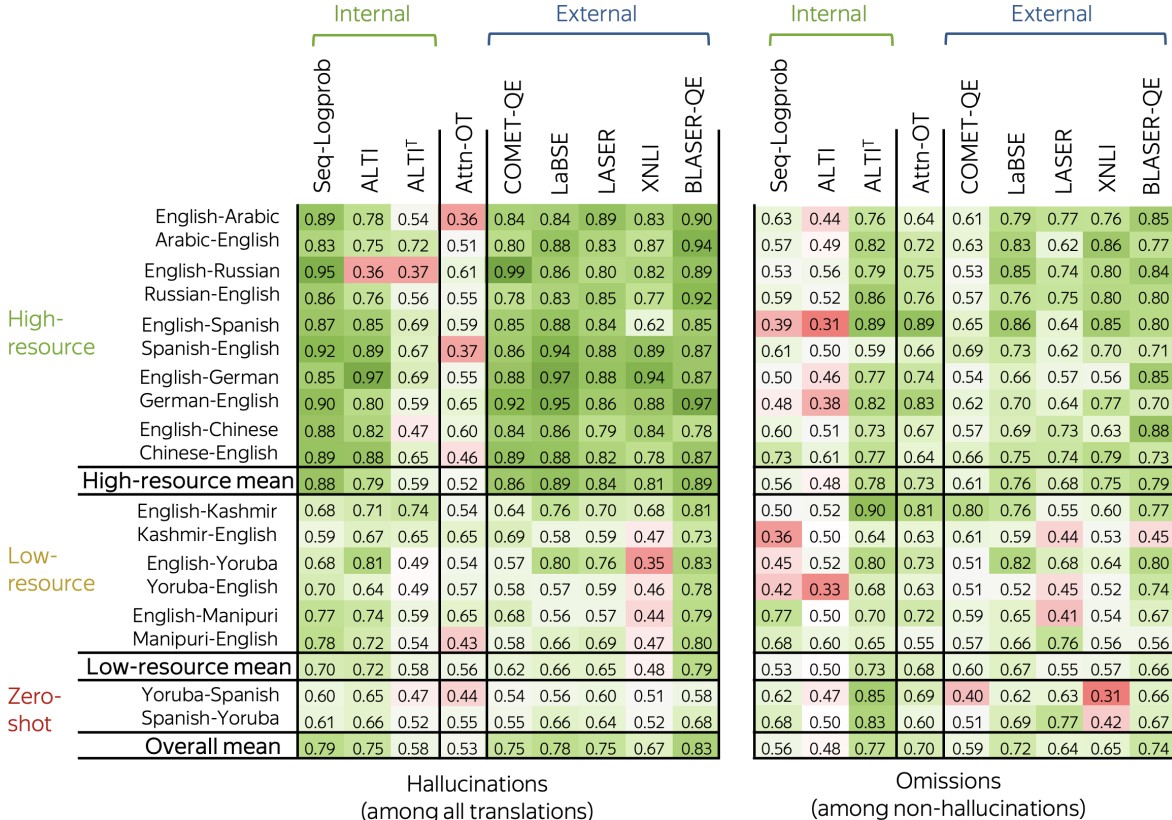

Figure 5: Results for sentence-level detection of hallucinations (left) and omissions (right).

gle language pair. This highlights the importance of our dataset enabling large-scale evaluation over several language pairs.

**BLASER-QE: a SOTA hallucination detector.** On average, BLASER-QE performs on par with the best hallucination detection methods high-resource directions, and outperforms them on low-resource directions. Apparently, fine-tuning massively multilingual sentence encoders to predict semantic similarity is a good recipe for hallucination detectors.

**Attention-based method: close to chance.** For hallucinations, Attn-OT detecting attention anomaly is an outlier and performs close to chance.[10][11] While previous work already showed that relying on attention patterns to make conclusions about translation quality can be fragile (Guerreiro et al., 2023), results with our dataset highlight

this even further. This points to a larger debate on the distinction between attention and attribution and the consequences of mistaking one for the other (Jain and Wallace, 2019; Serrano and Smith, 2019; Wiegreffe and Pinter, 2019; Bastings and Filippova, 2020). While Attn-OT was introduced as a way to evaluate detachment from the source sequence (Guerreiro et al., 2022), we see that implementing this intuition with attention instead of attribution (as in e.g. ALTI) leads to varying results: from high performance in Guerreiro et al. (2022) to near-random performance in our experiments.

**Omissions: internal ALTI[T] performs best.** For detecting omissions among non-hallucinations, the quality is generally worse than for hallucinations. The best method is ALTI[T] which confirms our intuition that if, according to token contributions, some source words are not used for translation, a translation is likely to omit relevant information. LaBSE, XNLI and BLASER-QE also perform well for high-resource languages but, similar to hallucination detection, are worse than internal methods for low-resource. Finally, while Attn-OT does not seem to identify hallucinations, it is sensible for omissions.

---

[10]We tried all the versions of the method from Guerreiro et al. (2022) as well as some additional modifications to improve its results. For the dataset from Guerreiro et al. (2022), we managed to reproduce their results. For out dataset, we show the best method variant in the main text and the rest (along with the implementation details) in Appendix B.

[11]For NLLB, poor performance of this method could be attributed to the overall large attention to the EOS token. We tried removing this token from the optimal transport computation, but this did not improve the results significantly.

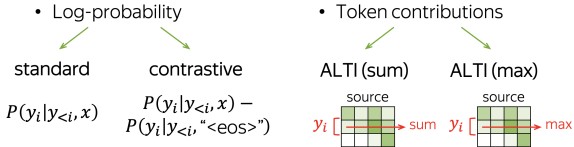

Figure 6: Token-level detection methods for hallucinations. For omissions, we swap the source and the target.

# 5 Word-Level Detection

In contrast to sentence-level detection, detecting pathologies at the word level received much less attention. In terms of both available data and detectors, previous attempts were rather limited (Zhou et al., 2021; Vamvas and Sennrich, 2022). Here, we want to facilitate future research in this direction.

**Detection tasks.** We define two detection tasks:

- *hallucination detection*: for each translation word, predict whether it is hallucinated;

- *omission detection*: for each source word, predict whether it is omitted from the translation.

**Segmentation.** We segment texts using Sacre-BLEU tokenizer (Post, 2018). For Chinese, it interprets each Chinese character as an individual word. For other languages, it applies regex-based tokenization based on spaces and punctuation.

**Evaluation methodology.** For these binary classification tasks, we use the ROC AUC score. Since models operate at the token level, we make predictions for tokens and not words. If a word is segmented into several tokens, we assign the worst score among its tokens (i.e., if one of a word's tokens is hallucinated, the entire word is hallucinated).

## 5.1 Detection Methods

To the best of our knowledge, there are no publicly available models for token-level detection of hallucinations or omissions that could be easily adapted to handle the language pairs in our dataset. Applying previous approach by Zhou et al. (2021), i.e. training a specialized model (or several language pair-specific models) on synthetic data, would be very demanding in terms of engineering and research effort to work well on diverse resource levels. Therefore, we suggest starting with internal methods and their combinations.

**Internal methods.** For internal methods, we rely on the same methods that were previously used: model log-probability and ALTI (Figure 6). We use

two types of log-probability: the standard and its difference with the unconditioned log-probability for the same token (i.e., when conditioning on an empty source sentence). Intuitively, the latter is one more way of measuring whether the model uses the source or relies more on its language model. For ALTI, we use both the total source contribution and the maximum contribution among the source tokens. The latter is high when the model is "focused" on a specific source token – this might be indicative of the prediction quality. For omissions, we use the same methods with swapped source and target sentences (i.e., ALTI turns into ALTI$^T$).

**Combination of methods.** Apart from the individual methods, we also consider their linear combinations. We use a logistic regression trained using 3-fold group-wise cross-validation (with sentence ids as groups). We train the same feature combination for all languages by maximizing the detection score on the pooled data.

## 5.2 Experimental Results

The results are shown in Figure 7. Overall, the methods we proposed are reasonable and perform much better than the random baseline.

**Token contributions perform best.** We see that for both hallucinations and omissions, token contributions coming from ALTI (or ALTI$^T$ for omissions) perform better than the log-probability coming from the model. Note that for hallucinations, this is different from the sentence-level results where Seq-Logprob outperformed ALTI.

**Contrastive vs standard log-probability.** Another interesting observation is that for hallucination detection in the high-resource setting, contrastive log-probability gives a better signal than the standard log-probability. This indicates that comparing explicitly the changes when dropping some information can be useful. This is not surprising: in a broad sense, our contrastive log-probability is a variant of erasure-based interpretation approaches (Zeiler and Fergus, 2014; Li et al., 2017; Kádár et al., 2017; Poerner et al., 2018; Li et al., 2019). In our case, the erased part is rather large, i.e. the whole source sentence. For such large segments, a similar idea previously appeared in context-aware machine translation when dropping the entire context sentence (Fernandes et al., 2021).

**The detectors are complementary.** Finally, we see that log-probability and token contributions are

| | Rand. | Logprob | | ALTI | | Comb. | |
|---|---|---|---|---|---|---|---|
| | | Stand. | Contr. | sum | max | | |
| High-resource mean | 0.50 | 0.73 | 0.78 | 0.87 | 0.71 | 0.93 | Hallucinations |
| Low-resource mean | 0.53 | 0.66 | 0.61 | 0.69 | 0.67 | 0.74 | |
| Overall mean | 0.51 | 0.70 | 0.70 | 0.78 | 0.68 | 0.84 | |
| High-resource mean | 0.51 | 0.84 | 0.79 | 0.86 | 0.83 | 0.92 | Omissions |
| Low-resource mean | 0.52 | 0.71 | 0.65 | 0.69 | 0.68 | 0.74 | |
| Overall mean | 0.51 | 0.78 | 0.73 | 0.78 | 0.76 | 0.83 | |

Figure 7: Word-level detection results.

complementary: for both hallucinations and omissions, combining the features leads to a noticeable improvement in detection quality.

## 6 Natural vs "Artificial" Pathologies

One of the difficulties when dealing with hallucinations (and, to a lesser extent, omissions) is that this is a rare phenomenon. Therefore, previous work often resorted to artificially amplifying the problem by applying various perturbations (Lee et al. (2019); Raunak et al. (2021), among others). However, it is not clear whether conclusions made based on this synthetic data would transfer to pathologies generated by a model in a natural setting.

In Appendix D, we compare performance of detection methods between two datasets: (i) our dataset with natural translations and (ii) translations generated with perturbed model. We find that data with translations generated under perturbation has to be used with caution, especially when evaluating pathology detection methods: the conclusions are likely to not transfer to the natural setting.

## 7 Discussion

We saw that some of the internal and external methods can detect hallucinations and omissions with the quality that is much better than nothing, much worse than perfect. But what are the cases in which methods perform well and what can be improved?

### 7.1 Sentence-Level Detection

Figure 8 shows manually selected examples of false and true positive and negative detection results.

**Flagging correct translations.** Examples 1-3 are correct translations, but some of them are flagged as pathological. Example 2 is flagged as an hallucination and omission, probably because the input ":::Jeez." is slang and has a wide range of potential meanings. Example 3 is flagged by Seq-Logprob: for the model, this translation may be "unlikely" because it is short and consists of a rare word.

**Difficult to detect pathologies.** Examples 4-6 show partial hallucinations and omissions that are difficult to detect, either because (in some sense) they resemble a correct translation, or because the translation indeed remains within the range of possible translations despite having these pathologies.

This raises a question: what does it really mean to have a hallucinated translation? While our sentence-level labels are fine-grained, the severity of a pathology is defined based on the number of omitted/hallucinated words rather than on the degree of semantic inadequacy of the pathology (similarly to e.g. Guerreiro et al. (2023)). This agrees with previous work noting that defining severity of translation errors is not straightforward (Graham et al., 2013; Licht et al., 2022; Guerreiro et al., 2023).

**Correctly detected pathologies.** Examples 7-11 show more severe hallucinations and omissions – these are detected correctly by at least a few of the considered methods. Many of these pathologies are produced for out-of-distribution inputs: conjunction of several sentences, typos, non-sentence texts (such as dates), and very short or incomplete sentences. Note that e.g. short sentences are non-typical for the NLLB training data. As we see, for such inputs the model is often not confident even for correct translations (see e.g. examples 2 and 3 – correct but short translations are flagged as pathological). This suggests that these errors might be alleviated by augmenting training data with similar (very short, multi-sentence, etc.) samples.

### 7.2 Word-Level Detection

In Appendix E.1, we also show examples of word-level detection and discuss the behavior of the detection methods. For example, we note that log-probability focuses on the beginnings of sentences while ALTI focuses on the endings.

## 8 Additional Related Work

Except for mentioned above work, previous work on hallucinations in machine translation largely avoided human annotation. To judge whether a translation is hallucinated, they relied on various heuristics or string-based automatic evaluation metrics (Lee et al. (2019); Berard et al. (2019); Müller and Sennrich (2021); Raunak et al. (2021)). These, however, were shown not to be indicative of hallucinations (Guerreiro et al., 2023), which highlights the importance of our human-annotated data. For omissions, previous work mostly focused on empty

| | Input | Output | Seq-Logprob | ALTI | ALTI$^T$ | LaBSE | |
|---|---|---|---|---|---|---|---|
| 1 | I'm glad Larry Grayson never edited Wikipedia, he would have had a dreadful time. | Me alegro de que Larry Grayson nunca editara Wikipedia, habría pasado un tiempo terrible. | -0.74 | 0.64 | 0.56 | 0.95 | true negative |
| 2 | :::Jeez. | Dios mío. | -1.99 | 0.58 | 0.36 | 0.48 | false positive |
| 3 | Kemet. | Kemet. | -2.76 | 0.69 | 0.69 | 1.00 | false positive by seq-logprob |
| 4 | If this group starts up... as a start, we should look at other advertising '<<<big guys>>>' and think about which sections we're like to expand/write about. - <<<Thanks, JG>>> | Si este grupo comienza... como comienzo, deberíamos mirar a otros "<<<granos>>>" publicitarios y pensar en qué secciones nos gusta ampliar / escribir sobre. | -0.72 | 0.60 | 0.52 | 0.86 | false negative hallucination |
| 5 | Former U.S. Speaker of the House Newt Gingrich came in second with 32 percent. | El ex presidente de la Cámara <<<de Representantes>>> de Estados Unidos, Newt Gingrich, quedó en segundo lugar con el 32 por ciento. | -0.39 | 0.64 | 0.87 | 0.96 | false negative hallucination |
| 6 | Most televisions are made <<<in a way>>> to please the general public. | La mayoría de los televisores están hechos para complacer al público en general. | -0.47 | 0.52 | 0.61 | 0.89 | false negative omission |
| 7 | Good <<<point>>>, I <<<just>>> assumed that they are the same <<<thing>>>. | Bueno, supuse que eran lo mismo. | -1.07 | 0.57 | 0.39 | 0.74 | true positive omission |
| 8 | However, the discovery of his tomb in 1922 made him a celebrity. <<<While many tombs of the past were robbed, this tomb was left virtually undisturbed>>>. | Sin embargo, el descubrimiento de su tumba en 1922 le convirtió en una celebridad. | -0.66 | 0.56 | 0.29 | 0.79 | true positive omission |
| 9 | :<<<Oh>>>... | <<<¿Por qué no lo haces?>>> | -2.18 | 0.37 | 0.67 | 0.29 | true positive hallucination |
| 10 | <<<z, , 2010-11-0104:56>>> | <<<Se trata de un proyecto de ley que se ha desarrollado en el ámbito de la seguridad social>>>. | -2.40 | 0.23 | 0.44 | 0.09 | true positive hallucination |
| 11 | Bye, | Hasta luego, <<<amigo>>>. | -2.58 | 0.58 | 1.02 | 0.63 | true positive hallucination |

Figure 8: Examples of successful and failed sentence-level detection for translations from English to Spanish.

translations (Stahlberg and Byrne, 2019; Vijayakumar et al., 2016) with some work using artificially created undertranslations (Vamvas and Sennrich, 2022). As we saw in Section 6, the latter is unlikely to be helpful when evaluating detection methods.

## 9 Conclusions

We present the first dataset with human-annotated hallucinations and omissions that satisfies several conditions: (i) it covers a broad range of languages with varying resource levels and scripts, (ii) the translations are generated naturally (i.e., without artificial perturbations), (iii) the model producing the translations is publicly available. Through our extensive experiments, we illustrate why each of these conditions is important. Additionally, we make several observations of individual importance. For example, for low-resource directions internal pathology detection methods perform better than most of the external ones, attention is very fragile when used to judge translation quality, among others. We believe our work opens the door for a reliable and accessible research on detecting and analyzing translation pathologies.

## 10 Limitations

Our experiments are reproducible, and our dataset together with the NLLB model can be widely used to evaluate progress on hallucinations/omissions detection and mitigation. However, all the annotated translations were generated with a single model, and the generalization to other models is yet to be verified.

The dataset is rather small and may not cover all possible hallucinations/omissions cases.

## 11 Ethical considerations

The annotations were provided by professionals and they were all paid a fair rate.

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

| Data source | FLORES | | | Comments | | |
| --- | --- | --- | --- | --- | --- | --- |
| Selection | U | B | W | U | B | W |
| eng_Latn-arb_Arab | 18 | 31 | 31 | 22 | 28 | 14 |
| arb_Arab-eng_Latn | 19 | 31 | 31 | 21 | 31 | 23 |
| eng_Latn-rus_Cyrl | 19 | 31 | 31 | 20 | 32 | 13 |
| rus_Cyrl-eng_Latn | 18 | 31 | 30 | 22 | 33 | 24 |
| eng_Latn-spa_Latn | 19 | 31 | 31 | 22 | 33 | 17 |
| spa_Latn-eng_Latn | 19 | 31 | 31 | 22 | 33 | 24 |
| eng_Latn-deu_Latn | 19 | 31 | 31 | 22 | 31 | 12 |
| deu_Latn-eng_Latn | 18 | 31 | 31 | 21 | 31 | 23 |
| eng_Latn-zho_Hans | 19 | 31 | 31 | 22 | 33 | 24 |
| zho_Hans-eng_Latn | 19 | 31 | 31 | 22 | 33 | 23 |
| eng_Latn-kas_Deva | 23 | 38 | 36 | 19 | 36 | 32 |
| kas_Deva-eng_Latn | 40 | 54 | 57 | 0 | 0 | 0 |
| eng_Latn-yor_Latn | 24 | 39 | 36 | 27 | 40 | 29 |
| yor_Latn-eng_Latn | 40 | 51 | 55 | 0 | 0 | 0 |
| eng_Latn-mni_Beng | 24 | 39 | 37 | 27 | 39 | 31 |
| mni_Beng-eng_Latn | 40 | 54 | 58 | 0 | 0 | 0 |
| yor_Latn-spa_Latn | 40 | 54 | 58 | 0 | 0 | 0 |
| spa_Latn-yor_Latn | 40 | 53 | 58 | 0 | 0 | 0 |

Table 1: Number of sentence pairs selected from each source with each method: uniform sampling U, biased sampling B, and selecting worst cases W.

Ghazvininejad. 2021. Detecting hallucinated content in conditional neural sequence generation. In *Findings of the Association for Computational Linguistics: ACL-IJCNLP 2021*, pages 1393–1404, Online. Association for Computational Linguistics.

Wenhao Zhu, Hongyi Liu, Qingxiu Dong, Jingjing Xu, Lingpeng Kong, Jiajun Chen, Lei Li, and Shujian Huang. 2023. Multilingual machine translation with large language models: Empirical results and analysis.

## A  Dataset Creation

**Selecting the data.** The 3 sampling strategies described in Section 2.2 were applied in different proportions, depending on what kind of data we had for a particular translation direction. Our released dataset has a field with the sampling strategy labels. The resulting proportions are reported in Table 1.

**Qualification tests.** The annotators recruited for this project were translators and reviewers who participated in FLORES translation (NLLB Team et al., 2022) or have other professional translation experience. Typically, these annotators are translators with at least two to three years of professional translation experience, usually with domain expertise in journalism, education, social media or marketing. Two annotators are recruited for each language. They are allowed to annotate our data only after passing a specifically developed qualification test. An annotator can fail the test no more

than once, in which case they are given an opportunity to receive a detailed feedback and re-do the test. If they do not achieve a passing score of 96% at the second attempt, the vendor is required to find a replacement. Once two annotators are qualified for a given language, one annotator performs the annotations, which are then reviewed by the second annotator.

Our qualification tests were developed for each of the language directions, and contained 15 items to annotate: 3 full hallucinations, 4 partial hallucinations, 2 word-level hallucinations, 5 mistranslations, and 1 incomprehensible sentence. The tests were found effective in identifying annotation quality issues before annotators annotate real data.

**Post-processing.** For each language, annotations were performed by one annotator and reviewed by another annotator. From the data, we discard the translations marked as incomprehensible along with the data with some issues (e.g. unbalanced brackets in the word-level annotations of hallucinations or omissions; word-level annotations that significantly differ from the initial input/output texts). After this filtering, we were left with 144 to 197 annotated sentence pairs per direction.

## B  Attention-based anomaly detection

**Reproducibility.** The Attn-OT sentence-level detection method that we use in Section 4 is our reproduction of the Wass-Combo method from Guerreiro et al. (2022). Their paper did not provide code and training data, so our implementation is not exact. For Wass-to-Unif, we obtained the same ROC AUC scores as Guerreiro et al. (2022) on their test set, but for Wass-to-Data and Wass-Combo, the AUC scores are 2% lower than in the original paper, probably due to the differences in selecting the reference data.

**Reference data for 18 directions.** To apply the Attn-OT method to our data, we created reference translations for each of the 18 translation directions by following steps:
- Sample 1M sentences for each language from the NLLB mined training data (NLLB Team et al., 2022);
- Translate them with the same settings as in Section 2.2;
- For each translation direction, drop the resulting sentence pairs that got into the worst 20% by any of the criteria: shortest-to-longest ratio

for source and translation texts, Seq-LogProb, and LASER3 cosine similarity score between source and translation.

After that, about 600K sentence pairs per direction are left as reference translations.

**Computing scores.** To compute attention distribution, we average the encoder-decoder attention maps for the last decoder layer over heads and over target tokens, just like Guerreiro et al. (2022). Our Attn-OT score is then computed with the same formula as for Wass-Combo in Guerreiro et al. (2022), with the only slight difference: $\tilde{s}_{wtu}$ is scaled by matching 1% and 99% quantiles of $s_{wtd}$, instead of min-max scaling, to improve computational stability. Along with this score, we also evaluate Wass-to-Unif and Wass-to-Data scores from Guerreiro et al. (2022), and their weighted average with weights inversely proportional to standard deviations: Wass-Mean.

**Dropping the EOS** We observed that in the setting above, Wass-to-Unif has nearly zero rank correlation with hallucination severity. After inspecting the attention maps, we found that about 75% of attention weight on most heads is distributed to the end-of-sentence token. This probably compensates for the fact that the order of magnitude of its encoder hidden state is an order of magnitude smaller than for other tokens, which aligns with the observations of Kobayashi et al. (2020). This makes a standard attention map highly non-uniform, and may obscure the more informative differences between the attention maps for different translations. To mitigate this fact, we computed the second version of all scores, with dropping the attention to the EOS token and renormalizing it so that the sum of attention to the other tokens equals 1 again.

**Evaluation** Table 2 reports ROC AUC scores for all OT-based detection methods, with and without including the EOS token. Removovg the EOS token improves the Wass-to-Unif and Wass-Mean scores, but slightly negatively affects Wass-to-Data and Wass-Combo scores. Whatever method we use, its performance for hallucination detection is not much better than chance.

## C   Detection of any pathology

Figure 9 reports scores for detecting hallucinations and omissions jointly. The scores are computed as percentage of correctly ranked pairs w.r.t. the worst

| Method | Hallucinations | Omissions |
|---|---|---|
| Wass-to-Unif | 0.49 | 0.43 |
| Wass-to-Data | 0.53 | 0.71 |
| Wass-Combo | 0.53 | 0.71 |
| Wass-Mean | 0.51 | 0.51 |
| Wass-to-Unif* | 0.55 | 0.65 |
| Wass-to-Data* | 0.51 | 0.69 |
| Wass-Combo* | 0.52 | 0.69 |
| Wass-Mean* | 0.55 | 0.67 |

Table 2: ROC AUC scores for detection of hallucinations and omissions with OT-based methods, averaged across tranlsation directions. The asterisk* denotes the scores based on the attention maps with the EOS token excluded.

of the hallucination level and omission levels for each sentence pair.

Qualitatively, the results are similar to those for hallucination detection: internal methods perform equally well for all translation directions, whereas external methods deteriorate for low-resource directions. For the joint detection of hallucinations and omissions, BLASER 2.0-QE outperforms all other methods both for high-resource and for low-resource directions.

## D   Natural vs Artificially Induced Pathologies

One of the difficulties when dealing with hallucinations (and, to a lesser extent, omissions) is that this is a rare phenomenon. Therefore, previous work often resorted to artificially amplifying the problem by applying various perturbations (Lee et al. (2019); Raunak et al. (2021), among others). However, it is not clear whether conclusions made based on this synthetic data would transfer to pathologies generated by a model in a natural setting. This is especially important when evaluating detection methods: for example, using internal workings of a model to detect its pathological behavior does not have to be helpful for pathologies the model did not generate "voluntarily". In this section, we compare performance of detection methods between two datasets: (i) our dataset with natural translations and (ii) translations generated under perturbation (annotated using the same protocol).

**Model perturbation.** To encourage the model to hallucinate or omit source information while still generating fluent text, we decrease the output acti-

Figure 9: Results for sentence-level detection of both hallucinations and omissions.

| | | Internal | | | | External | | | | |
| --- | --- | --- | --- | --- | --- | --- | --- | --- | --- | --- |
| | | Seq-Logprob | ALTI | ALTI$^T$ | Attn-OT | COMET-QE | LaBSE | LASER | XNLI | BLASER-QE |
| High-resource | English-Arabic | 0.71 | 0.55 | 0.70 | 0.56 | 0.69 | 0.82 | 0.81 | 0.79 | 0.87 |
| | Arabic-English | 0.75 | 0.68 | 0.64 | 0.54 | 0.78 | 0.83 | 0.74 | 0.79 | 0.86 |
| | English-Russian | 0.58 | 0.54 | 0.74 | 0.74 | 0.58 | 0.86 | 0.75 | 0.81 | 0.86 |
| | Russian-English | 0.63 | 0.54 | 0.74 | 0.76 | 0.73 | 0.80 | 0.73 | 0.81 | 0.85 |
| | English-Spanish | 0.72 | 0.64 | 0.73 | 0.67 | 0.67 | 0.81 | 0.80 | 0.80 | 0.83 |
| | Spanish-English | 0.76 | 0.69 | 0.61 | 0.64 | 0.73 | 0.81 | 0.78 | 0.77 | 0.82 |
| | English-German | 0.67 | 0.58 | 0.80 | 0.66 | 0.70 | 0.86 | 0.70 | 0.85 | 0.87 |
| | German-English | 0.64 | 0.66 | 0.74 | 0.67 | 0.68 | 0.79 | 0.70 | 0.71 | 0.81 |
| | English-Chinese | 0.59 | 0.52 | 0.84 | 0.79 | 0.74 | 0.89 | 0.74 | 0.79 | 0.87 |
| | Chinese-English | 0.84 | 0.78 | 0.72 | 0.53 | 0.81 | 0.84 | 0.80 | 0.81 | 0.83 |
| **High-resource mean** | | 0.69 | 0.62 | 0.72 | 0.65 | 0.71 | 0.83 | 0.76 | 0.79 | 0.85 |
| Low-resource | English-Kashmir | 0.67 | 0.70 | 0.77 | 0.57 | 0.67 | 0.77 | 0.69 | 0.68 | 0.82 |
| | Kashmir-English | 0.66 | 0.60 | 0.53 | 0.59 | 0.57 | 0.57 | 0.57 | 0.48 | 0.70 |
| | English-Yoruba | 0.58 | 0.69 | 0.64 | 0.64 | 0.55 | 0.82 | 0.73 | 0.48 | 0.83 |
| | Yoruba-English | 0.74 | 0.66 | 0.61 | 0.49 | 0.58 | 0.67 | 0.74 | 0.51 | 0.77 |
| | English-Manipuri | 0.56 | 0.65 | 0.68 | 0.68 | 0.70 | 0.58 | 0.58 | 0.50 | 0.79 |
| | Manipuri-English | 0.81 | 0.74 | 0.63 | 0.67 | 0.71 | 0.60 | 0.57 | 0.46 | 0.83 |
| **Low-resource mean** | | 0.67 | 0.67 | 0.65 | 0.61 | 0.63 | 0.67 | 0.65 | 0.52 | 0.79 |
| Zero-shot | Yoruba-Spanish | 0.61 | 0.63 | 0.54 | 0.48 | 0.51 | 0.58 | 0.63 | 0.47 | 0.61 |
| | Spanish-Yoruba | 0.66 | 0.62 | 0.63 | 0.57 | 0.53 | 0.69 | 0.70 | 0.49 | 0.70 |
| **Overall mean** | | 0.67 | 0.64 | 0.68 | 0.62 | 0.66 | 0.75 | 0.71 | 0.67 | 0.81 |

vations of all the encoder-decoder attention layers by a constant multiplier $\alpha$. Intuitively, this should imitate detachment from the source and increase hallucinatory rate. Indeed, translations generated this way are overall more pathological (see Figure 11). We use $\alpha = 0.3$ to match the average Seq-logprob of reference translations.

**"Artificial" data might not be informative.** Figure 10 shows sentence-level detection scores for different methods of hallucination and omission detection (average over all translation directions). We can see that perturbing the translation model introduces biases into the evaluation of detection methods. For example, for hallucination detection, Seq-Logprob outperforms ALTI on the natural dataset and loses on artificial. For omission detection, ALTI$^T$ is the best for the natural dataset, while XNLI is better on the artificial.

Figure 12 shows similar results, but with fractions of data downsampled in a way that the natural and perturbed data subsets have equal number of observations for each combination of pathology type, source dataset and translation direction. This is done to ensure that differences in detection per-

| | | Hall. | | Omis. | |
| --- | --- | --- | --- | --- | --- |
| | | Natural | Artificial | Natural | Artificial |
| Internal | Seq-Logprob | 0.79 | 0.72 | 0.56 | 0.60 |
| | ALTI | 0.74 | 0.83 | 0.48 | 0.51 |
| | ALTI$^T$ | 0.58 | 0.69 | 0.77 | 0.62 |
| | Attn-OT | 0.53 | 0.61 | 0.70 | 0.65 |
| External | COMET-QE | 0.75 | 0.71 | 0.59 | 0.58 |
| | LaBSE | 0.78 | 0.82 | 0.64 | 0.61 |
| | LASER | 0.75 | 0.76 | 0.64 | 0.59 |
| | XNLI | 0.67 | 0.63 | 0.65 | 0.69 |

Figure 10: Sentence-level detection scores for natural and artificial (generated under perturbation) pathologies.

formance to come from different distribution of pathology types. We can see that even for these curated subsets, the conclusions do not transfer from perturbed to natural pathologies.

Overall, we see that data with translations generated under perturbation has to be used with caution, especially when evaluating pathology detection methods: the conclusions are likely to not transfer to the natural setting.

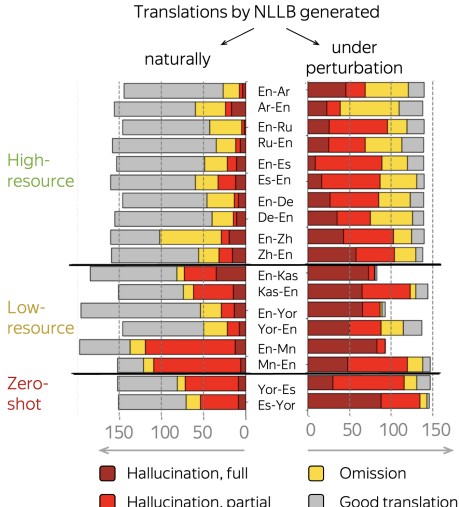

Figure 11: Annotation results: translations generated naturally vs under perturbation.

|  |  | Hall. |  | Omis. |  |
|---|---|---|---|---|---|
|  |  | Natural | Artificial | Natural | Artificial |
| Internal | Seq-Logprob | 0.81 | 0.76 | 0.54 | 0.59 |
|  | ALTI | 0.77 | 0.83 | 0.47 | 0.51 |
|  | ALTI$^T$ | 0.58 | 0.65 | 0.76 | 0.61 |
|  | Attn-OT | 0.50 | 0.61 | 0.65 | 0.64 |
| External | COMET-QE | 0.72 | 0.72 | 0.50 | 0.58 |
|  | LaBSE | 0.81 | 0.83 | 0.68 | 0.60 |
|  | LASER | 0.74 | 0.73 | 0.61 | 0.59 |
|  | XNLI | 0.66 | 0.68 | 0.59 | 0.67 |

Figure 12: Word-level detection results.

# E Discussion

## E.1 Word-Level Detection

Figure 13 shows examples of word-level detection.

**Logprob focuses on the beginnings.** We notice that log-probability focuses more on the beginning of a word or a sentence. This makes sense: model uncertainty in prediction is generally higher when beginning generation.

**ALTI focuses on the endings.** Differently, token contributions focus on word endings. This is again expected: when generating a token that completes a word, source contribution is likely to be lower than for the other tokens. However, the predictions are still very reasonable – for the last three examples, ALTI detects omissions and hallucinations more confidently than log-probability.

Finally, we see that feature-based combination of the methods leads to more refined results.

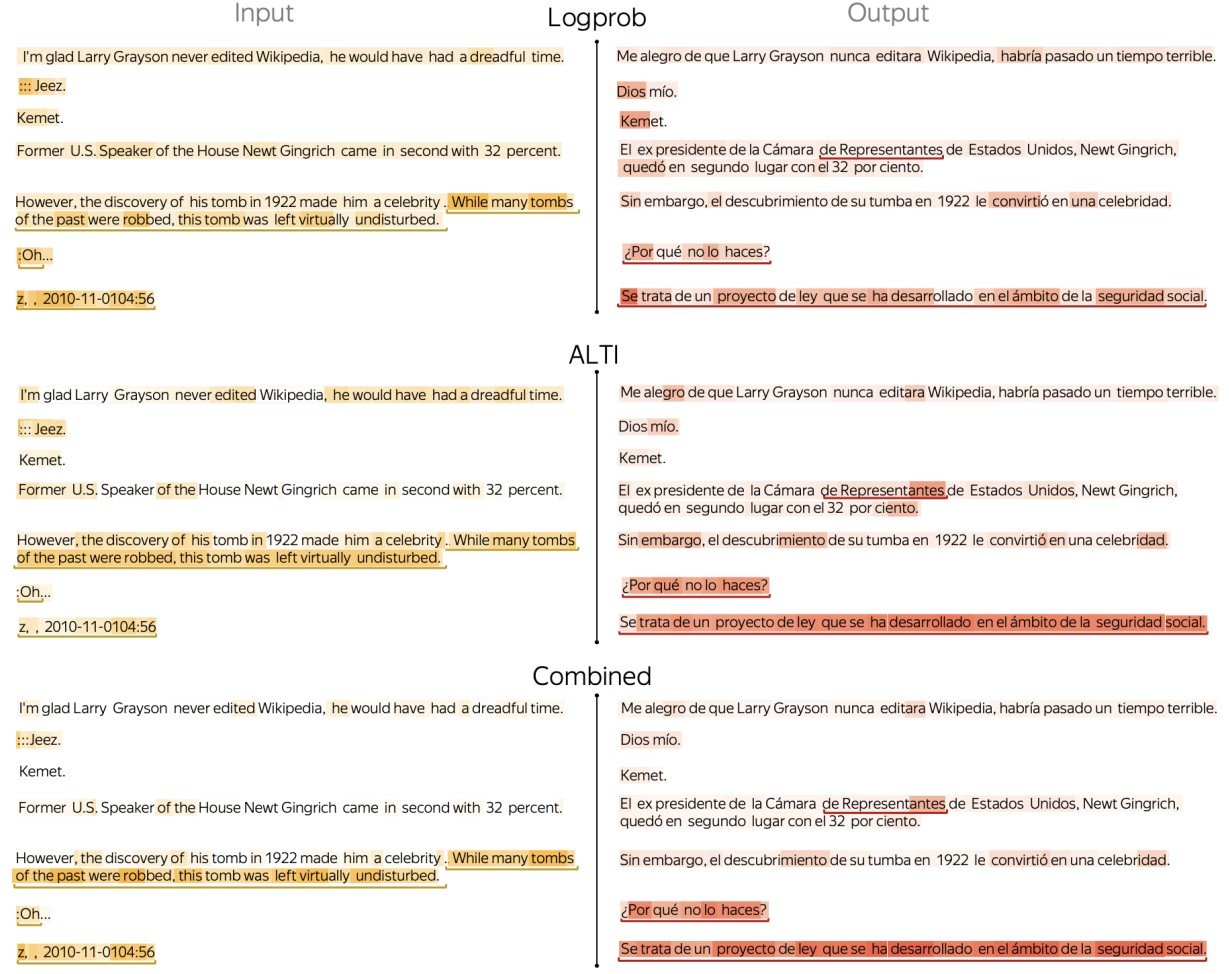

Figure 13: Examples of word-level detection for translations from English to Spanish. Hallucinated and omitted fragments are underlined.