# OpenReview forum: "HalOmi: A Manually Annotated Benchmark for Multilingual Hallucination and Omission Detection in Machine Translation"
_EMNLP/2023/Conference — EMNLP 2023 Main_

### Official Review · Reviewer_XBnn · 2023-07-25

**Soundness:** 4

**Excitement:**

4: Strong: This paper deepens the understanding of some phenomenon or lowers the barriers to an existing research direction.

**Paper Topic And Main Contributions:**

The paper presents a new dataset of machine translation hallucination and omission errors, including sentence-level and token-level annotations for eighteen translation directions involving nine languages. They evaluate multiple hallucination and omission detection methods on the dataset, finding that existing detection methods perform much better on high-resource language pairs. Sequence log-probability is best for sentence-level hallucination detection, and token attribution methods are best for omission detection or token-level detection.

**Questions For The Authors:**

A: How were translators recruited, particularly for the low-resource languages?

B: How might function words and languages' morphology affect token-level hallucination/omission annotations and evaluation scores? E.g. words like "to" and "the" behave very differently across languages (e.g. sometimes as affixes to content words, or sometimes as standalone words). In many cases, it seems unclear how to annotate function words given the provided hallucination/omission guidelines. This would affect token-level detection scores.

**Reasons To Accept:**

The dataset is a useful resource for future work on hallucinations and omissions in machine translation systems. It is particularly interesting to see the differences in existing detection methods' performance between low and high resource language pairs.

**Reasons To Reject:**

The main text is longer than 8 pages.

The dataset only contains hallucination and coverage errors for the 600M distilled NLLB model.

The limitations section is quite short; it could mention other limitations, e.g. ambiguity in the definition of hallucinations and omissions, difficulties in word-level annotation (e.g. depending on different languages' morphology), focusing on pairs to/from English, etc.

**Reproducibility:**

3: Could reproduce the results with some difficulty. The settings of parameters are underspecified or subjectively determined; the training/evaluation data are not widely available.

**Reviewer Confidence:**

3: Pretty sure, but there's a chance I missed something. Although I have a good feel for this area in general, I did not carefully check the paper's details, e.g., the math, experimental design, or novelty.

---

> ### Author Rebuttal · Authors · 2023-08-28
>
> Thanks for your comments!
>
> QA: The vendor has a large pool of resources in their community database and the vendor’s Community Management team would screen for these linguists to find those who might be a good match for the project needs.
>
> QB: We are not sure we understand the question, but it could be that the reviewer may allude to the fact that in some directions, the target language uses function words that the source language doesn't use. In that case, the target language would have more "tokens" than the source language, which could be confused with a hallucination phenomenon. This being said, annotators were advised to avoid jumping to conclusions whenever they identified erroneous target words. We do not believe that this is the case but do not have specific data on the topic at this point.

---

### Official Review · Reviewer_Kt9P · 2023-08-02

**Soundness:** 4

**Excitement:**

4: Strong: This paper deepens the understanding of some phenomenon or lowers the barriers to an existing research direction.

**Paper Topic And Main Contributions:**

This paper is about developing a new dataset for detecting hallucinations and omissions in machine translation. The dataset is quite general; it contains 18 low and high resourced language pair directions. The dataset is quite useful, as it provides a way for MT systems to be trained to detect such pathological translation mistakes. The paper shows how existing system which were developed for detecting such pathologies, perform on the new dataset, with interesting results.

**Questions For The Authors:**

I am not sure I totally understand what is the background of the annotators. The authors do say something about it, but I think it should be discussed with more details.

**Reasons To Accept:**

This paper should be accepted for EMNLP as it provides a very useful dataset for the research community, which was not available before. I find the annotation guidelines clear and simple. However, I do think more annotation examples should have been provided. The existing examples in Figure 3 are probably not enough.

**Reasons To Reject:**

1. The paper should be accepted. However, it does go beyond the 8 pages requirement (the Conclusions section crosses over to the 9th page), which is not allowed in this setting.
2. The translations annotated in this project, were all generated by the same model. I recommend adding another MT model to make sure the data is not biased.

**Reproducibility:**

3: Could reproduce the results with some difficulty. The settings of parameters are underspecified or subjectively determined; the training/evaluation data are not widely available.

**Reviewer Confidence:**

5: Positive that my evaluation is correct. I read the paper very carefully and I am very familiar with related work.

---

> ### Author Rebuttal · Authors · 2023-08-28
>
> Thanks for your comments. In the current paper, we focused on using a single model for all translation directions, to facilitate comparison between them and evaluation of internal detection methods. Moreover, the only publicly available model family covering all the languages is NLLB, and adding another NLLB model (that shares training data and some hyperparemeters with NLLB-600M) would probably preserve a lot of model-based bias in the data.
>
> Given the difficulty and time of the annotation process, we will not be able to add translations of another model in the final version of the current paper. But we are planning to address this in the future research.
>
>
>
> Regarding your question about annotations: Annotators who supported this work are professional translators with at least 2-3 years of professional translation experience, usually with domain expertise in journalism, education, social media and marketing.

---

### Official Review · Reviewer_TKvD · 2023-08-05

**Soundness:** 4

**Excitement:**

4: Strong: This paper deepens the understanding of some phenomenon or lowers the barriers to an existing research direction.

**Paper Topic And Main Contributions:**

The paper presents a multilingual dataset and benchmark for the detection of hallucinations and omissions in machine translation. The dataset consists of sentence-level and token-level manual annotations for 18 translation directions providing a valuable novel and comprehensive benchmark. The paper also includes results from known detection methods and also proposes a new metric for the reliable detection of omissions.

**Questions For The Authors:**

Did you measure annotation agreement and did you consider to resolve or keep annotation variation?
The choice of NLLB 600: selecting the smallest model, is this because of inference costs or better chance of finding omissions and hallucinations?
How did you combine the 3 sampling strategies? Did you take equal amounts from each of them in the final dataset? Is it interesting to look at the impact of sampling strategy somehow?

**Reasons To Accept:**

This is a very much needed dataset that will enable systematic tests of detection methods of hallucinations in machine translations. The experiments also show that a good language coverage is necessary to draw conclusions that generalize. The manual annotation effort is important to avoid artificial settings that do not reflect the real nature of the problem.

The paper is very well written, clearly presented and the dataset and detection methods are clearly described and discussed. The analyses and discussion contributes valuable information to this important topic and the resources are reusable and very valuable.

**Reasons To Reject:**

There is almost no information about the annotation process and the work of the annotators. I would like to see some discussions about the difficulty of the annotation work. I can imagine that there i a lot of variation in opinions on hallucination and I would like to know how the authors dealt with discrepancies and disagreements.

**Reproducibility:**

4: Could mostly reproduce the results, but there may be some variation because of sample variance or minor variations in their interpretation of the protocol or method.

**Reviewer Confidence:**

4: Quite sure. I tried to check the important points carefully. It's unlikely, though conceivable, that I missed something that should affect my ratings.

---

> ### Author Rebuttal · Authors · 2023-08-28
>
> Thanks for your comments!
>
> Regarding the annotator details: for each language, 2 annotators were recruited by a language service provider. Once recruited, both annotators were given a qualification test. To qualify to work on the project, annotators had to pass the test (passing grade = 96%).
>
> Q1: For each language, annotations were performed by one annotator and reviewed by another annotator.
>
> Q2: We chose NLLB-200-600M model for two reasons: (1) it is the most popular version of the model (by number of likes on HuggingFace), and (2) it would probably produce the largest number of hallucinations.
>
> Q3-Q5: the 3 sampling strategies were applied in different proportions, depending on what kind of data we had for a particular direction. Our released dataset will have the sampling strategy label and we will report the final proportions in the final version of the paper.

---

### Meta-Review · Area_Chair_cPox · 2023-09-15

**Recommendation:** 4

**Metareview:**

The paper presents a multilingual dataset and benchmark for the detection of hallucinations and omissions in machine translation.
The dataset is quite general; it contains 18 low and high resourced language pair directions. The dataset is quite useful, as it provides a way for MT systems to be trained to detect such pathological translation mistakes. The authors evaluate multiple hallucination and omission detection methods.  The paper shows how existing system which were developed for detecting such pathologies, perform on the new dataset, with interesting results.

---

### Decision · Program_Chairs · 2023-10-07

**Decision:**

Accept-Main

**Comment:**

The paper presents a multilingual dataset and benchmark for the detection of hallucinations and omissions in machine translation.
The dataset is quite general; it contains 18 low and high resourced language pair directions. The dataset is quite useful, as it provides a way for MT systems to be trained to detect such pathological translation mistakes. The authors evaluate multiple hallucination and omission detection methods.  The paper shows how existing system which were developed for detecting such pathologies, perform on the new dataset, with interesting results.